
**Introduction to the special issue "Venice flooding and sea level: past evolution, present issues and**
**future projections"**
Piero Lionello[1], Robert J. Nicholls[2], Georg Umgiesser[3], Davide Zanchettin[4]
[1]Università del Salento, Dept. of Biological and Environmental Sciences and Technologies, Centro
Ecotekne Pal. M - S.P. 6, Lecce Monteroni, Italy
[2]Tyndall Centre for Climate Change Research, University of East Anglia. Norwich NR4 7TJ, United
Kingdom
[3]CNR - National Research Council of Italy, ISMAR - Marine Sciences Institute, Castello 2737/F, 30122
Venezia, Italy
[4]University Ca' Foscari of Venice, Dept. of Environmental Sciences, Informatics and Statistics, Via
Torino 155, 30172 Mestre, Italy
correspondence to: Piero Lionello (piero.lionello@unisalento.it)

**Abstract.**
Venice is an iconic place and a paradigm of a huge historical and cultural value at risk. The frequency
of flooding of the city centre has dramatically increased in recent decades and this threat is expected to
continue to grow and even accelerate through this century. This special issue collects three review
papers addressing different and complementary aspects of the hazards causing the flooding of Venice:
(1) the relative sea level rise, (2) the occurrence of extreme sea levels, and (3) the flood prediction. It
emerges that the effect of compound events poses critical challenges to the forecast of floods,
particularly from the perspective of effectively operating the new MoSE mobile barriers. Two strands
of research are needed in the future. Firstly, there is a need to better understand and reduce the
uncertainty on the future evolution of relative sea level and its extremes at Venice. However, uncertainty
might not be substantially reduced in the near future, reflecting uncertain anthropogenic emissions and
structural model features. Hence, complementary adaptive planning strategies appropriate for
conditions of uncertainty should be explored and developed in the future.

**1. Motivation**
The city of Venice and its lagoon represent great historic, ecologic, and economic interest that is known
around the world. In 1987, Venice and its lagoon was recognised as a UNESCO World Heritage Site
based on six criteria of outstanding cultural, environmental and landscape universal value encompassing
the historical and artistic relevance of Venice and the exemplarity of the Venice Lagoon ecosystem
(https://whc.unesco.org/en/list/394/). In UNESCO's words, Venice symbolizes "*the people's victorious*
*struggle against the elements as they managed to master a hostile nature*" and its semi-lacustral habitat
"*has become vulnerable as a result of irreversible natural and climate changes*".
The history and the very essence of Venice are tightly intertwined with the sea, which has represented
a source of resources and wealth, and a natural defence system against enemies. However, it has always
brought the threat of floods (Enzi and Camuffo, 1995). This hazard has been exacerbated by ongoing




relative sea level (RSL) rise over the last 60 years (Lionello et al., 2012), posing serious and growing
threats to the city of Venice and its lagoon. In fact, the recurrent floods that afflict Venice, locally
referred to as "aqua alta", are the best known and debated symptom of the frailty of the Venetian lagoon
system. The Venetian RSL has risen more than 30 cm in the last 150 years due to mean sea level rise
and sinking of the ground by natural and anthropogenic subsidence (Zanchettin et al. 2020) and this has
increased the frequency of floods (Lionello et a., 2020). Venice's central St. Mark's Square is
approximately 55 cm above the present mean RSL and nowadays even a moderate surge can flood it, if
it coincides with a maximum of the astronomical tide, whose amplitude is about 50 cm. The average
ground level of the city is approximately 80 cm and the critical 120 cm threshold leading to the flood
of a large fraction of it has been reached  by 40 events in the last decade 2010-2019[1].
The dramatic surge of 4 November 1966 showed unequivocally the need for counteracting an increasing
hazard level. The event reached the highest ever recorded RSL (194 cm) and persisted over 110 cm for
22 hours (see Lionello et al., 2020, De Zolt et al., 2006, Cavaleri et al., 2010). Figure 1 shows the
flooding of the central monumental area at a time close to the peak of the 6 November 1966 event. In
1973, the Italian government established a legal framework, the Special Law for Venice, establishing
objectives, responsibilities, regulations, actions and funding allocations to respond to the need of
safeguarding Venice and its lagoon. The solution finally approved by the Italian government is a system
of large mobile barriers (MoSE, Modulo Sperimentale Elettromeccanico) at the three lagoon inlets.
Barriers will be raised only during storm surges, closing the lagoon inlets and preventing sea level
exceeding the safeguard level within the lagoon, while under normal conditions, they lay on the bottom
of the lagoon inlets, allowing ship traffic and water exchange between the lagoon and the Adriatic Sea.
After a long planning phase, construction started in 2003 and it is foreseen that the barriers will be fully
operational at the end of 2021. The recent event on 12 November 2019 (Cavaleri et al., 2020, Ferrarin
et al.,2020), which has been the second highest RSL (189 cm) ever measured in Venice by a tide gauge,
has dramatically reconfirmed the need for an adequate defence system. November 2019 was the worst
month since the beginning of the local sea-level records for excessive high waters, with 15 high tides
exceeding 110 cm and 4 events above 140 cm. At present, it is possible to operate MoSE for exceptional
events, as was done on 3 October 2020, when, for the first time, the lagoon has been cut off under real
conditions from the Adriatic Sea. While the peak sea level during the event was 130 cm in the Adriatic,
in the lagoon and at St. Mark's Square, it was kept at 70 cm and flooding was avoided. Figure 2 shows
an aerial view of the barriers blocking the lagoon inlets and the time series of the RLS outside
("Piattaforma") and inside the lagoon ("Punta Salute Canal Grande" represents the tide gauge
commonly used as reference for the sea level in the city center).
A large scientific literature considers the factors leading to the flooding of Venice, predicting the timing
and intensity of the events and describing changes in their frequency and intensity under future global
warming scenarios. This special issue aims to critically review the current understanding of the Venice
flooding phenomenon. It considers the meteorological and climatic factors producing "aqua alta", its
prediction, and its historical and expected future variations under a globally changing climate. The
synthesis is oriented toward clarifying consolidated knowledge, highlighting gaps in knowledge, and
identifying major opportunities for progress.
This special issue comprises three review papers addressing three different and complementary aspects
of the hazards causing the flood of Venice. Zanchettin et al. (2020) consider the Venetian RSL evolution
on multiple time scales and the factors determining it. Umgiesser et al. (2020) describe the tools that

---

[1] https://www.comune.venezia.it/it/content/grafici-e-statistiche (data extracted on 27 October 2020)




have been developed and are currently used for the prediction of the sea level events. Lionello et al.
(2020) describe the factors leading to extreme sea-level events, their past evolution, and expected future
trends under a climate change perspective. The outcomes of these papers provide a thorough critical
review of the scientific literature. It is, hence, a basis for the assessment of present and future risks and
helps to define the requirements of the adaptation strategies needed for Venice over the 21[st] century.
This editorial provides an introduction to these three reviews. It shortly provides general  background
information by describing the geographical and historical setting (section 2) and the phenomenology of
surges and high water levels (section 3). Section 4 describes the overall key findings produced by the
three reviews. Implications for future flooding and its management are addressed in the conclusive
section 5.
**2.  Geographical and historical setting**
The Venice Lagoon covers about 550 km[2] along about 50 km of low-lying coast within the easternmost
boundary of the Po Plain, and is connected to the northern Adriatic Sea through three tidal inlets: Lido,
Malamocco and Chioggia (Figure 3). The historical city center is located in the centre of the lagoon and
is built on piles at low elevation.
The Venice Lagoon is governed by a fragile equilibrium, which has been artificially preserved over the
centuries by contrasting the natural evolution of this transitional area that is driven by coastal dynamics
via geomorphological (e.g., erosion), chemical (e.g., salinification), biological and ecological (e.g., loss
of wetlands and other ecotopes) changes. Since the 15[th] Century Venetians have engaged in an enduring
struggle against sedimentation in the lagoon, mainly by diverting away - with variable degrees of
success - the major rivers Adige, Bacchiglione, Brenta, Sile, and Piave and their sediment supply, hence
altering the morphology of the alluvial plain and the coastal margins. Bondesan and Furlanetto (2012)
provide a recent assessment, based on historical cartography analysis, of the artificial fluvial diversions
performed during the 16[th] and 17[th] Centuries. More recent works in the 19[th] and 20[th] century include
deepening of existing channels within the lagoon, excavation of the "Canale dei Petroli" (Oils Channel)
and construction of breakwaters at the lagoon's mouths to allow modern ships to reach the ports of
Giudecca and Marittima in the historic city and, more recently Porto Marghera.
The tidal regime in the Northern Adriatic Sea, and therefore in the Venice Lagoon, is dominated by the
semi-diurnal components with a tidal range of more than 1m at spring tide (Ferrarin et al., 2015).
Hydrodynamics linked to tidal exchange are critical for the great ecological variety and biodiversity of
the Venice Lagoon, with habitats ranging from tidal flats, marshlands, channels and canals, inlets and
tidal deltas with strong hydrodynamics and tidal renewals.
Changes in RSL  may critically compromise the ecosystem functionality by inducing morphodynamic
changes that alter the ecological vocation of such areas (Zanchettin et al., 2007). Several studies show
that the increase of extreme floods since the mid-20th century is explained by RSL rise (Lionello et
al.,2012, Lionello et al. 2020). Further, future sea level rise might dramatically increase both the
frequency of high sea level events and resulting floods, as well as increasing the duration and extent of
flooding (Lionello et al. 2020). This further leads to the need to understand the historical context of sea-
level change in Venice, and consider its prognosis.
**3.  Characteristics of surges and high water levels**


Storm surges in Venice are caused by a combination of various physical processes, mainly triggered by
the water level in the neighbouring Adriatic Sea. The main components are the stress from south-east
winds (Sirocco), pushing the water versus the north-western end of the Adriatic Sea, and a low
atmospheric pressure that increases the mean sea level by one centimetre per millibar of pressure
decrease. To these two meteorological processes, the contribution of the regular tides has to be
considered, which adds about 50 cm during a spring tide (Ferrarin et al., 2015).
Another flood process is seiches, i.e., free oscillations of the Adriatic Sea triggered by wind setup. These
seiches, which have a period of around 23 hours, very close to the diurnal frequency of the tides, overlay
the meteorological and tidal processes and may cause flooding even if the main meteorological
conditions have calmed down (Bajo et al., 2019).
If all these forcings are in phase with each other the "aqua alta" phenomenon results. The positive water
level anomaly in the Adriatic Sea enters the lagoon nearly undisturbed through the deep inlets (8-13
meters) and then reaches and floods the city center of Venice (Umgiesser et al., 2004; 2020). While
there might be some local differences due to wave setup outside the inlets and wind stress inside the
lagoon, the water level in the city of Venice closely follows the water level outside the lagoon.

## 4. Key insights from the papers

The sea level forecast (Umgiesser et al., 2020) has paramount importance, because it is needed by civil
protection for flood warnings and by the consortium that operates the mobile barriers (MoSE), which
are in a pre-operational phase at the inlets. Considering the operativity of MoSE, a reliable forecast
should be able to satisfy the requirements of the different stakeholders, especially in terms of the
forecast range and error statistics (Umgiesser, 2000). The present plan is to operate the barriers and to
close the lagoon on the basis of the forecast water level, wind and rain only few hours before the event.
The port authority is particularly sensitive to unnecessary closures, which produce unmotivated
economic losses by limiting the port operations and to anticipate (in the range one to two days) the
decision to close, which facilitate proper management of the ingoing and outgoing ship traffic.
Residents, shopkeepers and most commercial activities in Venice would support a more conservative
approach that reduces to a minimum the risk of flood damages to goods and properties. Therefore, the
port authority is interested to avoid false alarms, while other stakeholders are worried about missing
closures. Tourist activities would in general be concerned by cancellations of reservations and visits
that may be caused by an excessive water level forecast.
An operational forecasting system has been in place for the last 40 years, but further developments are
needed to match the requests of stakeholders and the requirements for operating MoSE. Lack of
accuracy in the forecast of the compound event that led to the exceptional sea level maximum on 12
November 2019, produced a severe underestimate (up to 45 cm) of the maximum event height by all
available forecast systems (Ferrarin et al., submitted). Therefore other developments are needed, like
the use of ensemble methods, assimilation of real time data, and the exploitation of multi model
approaches (Umgiesser et al., 2000). Implementing these features in the forecasting systems can (and
should) be done to guarantee an improved and adequate water level forecast in the near future.
The important potential role of compound events for causing extreme sea levels emerges clearly from
Lionello et al. (2020). Many past studies concentrated on the storm surge contribution (which was the
determinant for the 4 November 1966 event) and on the need for a precise prediction of its timing in
relation to the phase of astronomical tide and of pre-existing seiches. However, the presence of other
factors, namely planetary atmospheric wave surges and meteotsunamis can determine extreme sea-level


events when they act constructively (Lionello et al., 2020), as was apparent in the recent 12 November
2019 event. This poses a great challenge to the prediction of extreme sea levels (Umgiesser et al., 2020)
and the management of MoSE. Further, historic floods show large interdecadal and interannual
fluctuations, whose dynamics are not sufficiently understood (Lionello et al., 2020), preventing reliable
seasonal predictions.
RSL rise is the factor that has produced the past increase of the Venice flood frequency. Zanchettin et
al. (2020) provides, by comparing available studies, a RSL trend in Venice of approximately 2.5
mm/year during the 20$^{th}$ century, caused in approximately equal parts by land subsidence and mean sea
level (MSL) rise. Lionello et al. (2020) shows that increased frequency of floods is attributed to such
RSL rise, with no robust evidence of intensification of the meteorological conditions associated with
extreme sea levels. Figure 4 summarizes these results by showing the RSL rise in Venice and the
corresponding increase of frequency of sea-level maxima above 120 cm with respect to the standard
reference RSL level.
Uncertainty in future greenhouse gas emissions (largely depending on governmental and societal
decisions) and structural modeling uncertainties (particularly in relation to the melting of the large
Greenland and Antarctic ice sheets) lead to a wide range of possible future sea level rise scenarios
(Zanchettin et al., 2020). Figure 4 shows that the past MSL in Venice closely follows the MSL trend of
the Subpolar North Atlantic. Differences between these two time series consists of inter-annual and
inter-decadal sea level fluctuations in the North Adriatic, with no sustained different trends. This
supports studies showing that future sub-regional deviations play a minor role and can be estimated to
be of the order of ±10 cm (Zanchettin et al., 2020). Figure 5 shows a RSL rise range from 40 to 110 cm
at the end of the 21$^{st}$ century (with a wider 17 to 120 centimeter range, accounting for uncertainty). This
could grow to above 180 cm if an unlikely, but plausible high-end scenario is realised. These values are
obtained considering regional analysis of future RSL (Thiéblemont et al., 2019), integrated by
accounting for centennial natural vertical land movement occurring at the past rate (Zanchettin et al.,
2020) and adding a further 10 cm uncertainty caused by sub-regional deviations from the Subpolar
North Atlantic sea level.
Future RSL rise will be the key factor determining the future duration of extreme sea levels above the
safeguard thresholds, which correspond to the duration of the closures of the inlets by the MoSE mobile
barriers. Figure 5 reports the RSL thresholds for the closures based on the consensus between  Lionello
(2012) and Umgiesser (2020). It shows that closing of the inlets for three weeks per year is likely as
soon as the 2040s and the period of closure will be growing at a rate controlled by RSL rise. Longer
closure periods (e.g., 2 months per year) might be anticipated in the late 2050's under a high-end
emission scenario (RCP8.5) or at the end of this century for a moderate  scenario (RCP2.6).
RSL rise will also be the key factor responsible for the future increase of extreme sea-level frequency
and height, while the reduction of intensity of meteorological events and changes of tidal regimes will
play a secondary role (Lionello et al., 2020). In the case of a high-emission scenario (RCP8.5) the
magnitude of 1-in-100 year sea-level events at the North Adriatic coast is projected to increase up to
65% and 160% in 2050 and 2100, respectively, with respect to the present value, continuing to increase
thereafter.
**5.   Implications for future flooding and its management**
The insights from the papers have important implications for the future occurrence of flooding in Venice
and its management. They demonstrate that RSL rise has been and will continue to be the main driver





of increasing extreme sea levels and increasing flood potential. Projected future RSL rise is the product
of local changes due to subsidence and regional and global trends linked to human-induced climate
change. Natural background subsidence (up to around 1 mm/year) due to enduring long-term geological
trends apparent over many centuries and longer is inevitable in Venice. Importantly, however, most
subsidence in the last 100 years was due to human agency (largely groundwater withdrawal). Since the
1970s, good regulation and provision of alternative sources of water for industrial, agricultural and civil
use have avoided such subsidence. It is important that these successful regulations to control human-
induced subsidence continue to be enforced in the future. Efforts could even be strengthened further, as
some localised subsidence can still be measured linked to construction works and related activities (Tosi
et al., 2018). Costs, benefits and practicality of the required regulations might be considered in the
context of building regulations and permitting. Therefore, future human-induced contributions to local
subsidence can be controlled based on historic experience and awareness.
In contrast, most ongoing and projected future climate-induced sea-level rise is a result of global actions
concerning greenhouse gas emissions and resulting temperature rise. It is therefore of paramount
importance to identify and support collective global actions to reduce such emissions, especially the
Paris Agreement. It is also important for Venice, as in other coastal jurisdictions around the world, to
stay aware of future expectations about sea-level rise and plan accordingly. The regular assessments of
the Intergovernmental Panel on Climate Change (IPCC) are especially important in this regard, with
the Sixth Assessment being expected in 2021. Currently the world is heading towards emissions most
comparable to RCP4.5, rather than RCP8.5 (Hausfather and Peters, 2020). With further reductions,
emissions close to RCP2.6 (following the Paris Agreement) is a plausible albeit challenging target to
achieve. However, the recent IPCC Special Report on the Cryosphere and Oceans (Oppenheimer et al.,
2019) has emphasised the fundamental point that stabilizing temperature does not stabilize sea level,
but rather, the rate of sea-level rise.
Hence, some RSL rise is inevitable for Venice and extreme sea levels and flood potential will grow:
uncertainty concerns only the rate of this increase. Significant aspects of this uncertainty relate to future
emissions, the response of the Greenland and Antarctic ice sheets to global temperature rise and future
subsidence of the Venice Lagoon. It is important to remember that Venice has adapted to RSL rise
through its more than 1,000 year history. Hence, the adaptation actions since the 1966 floods,
comprising both local adaptation (by raising parts of the historical center), and large-scale adaptation
for the whole lagoon (the construction of the MoSE barriers) continue this tradition. With the MoSE
barriers being fully commissioned in 2021 the risks of flooding in Venice will be greatly reduced.
However, as RSL is still rising and is projected to rise beyond the 21st Century even with the Paris
Agreement being fully followed, ultimately even this new world-class adaptation system will be
challenged. The critical question is when a new adaptation strategy will be required, being aware that it
might happen as soon as the 2050s considering the uncertainty of future RSL scenarios, or maybe much
later. This suggests that experience from long-term planning for sea-level rise in locations such as
London (Ranger et al., 2013) and drawing on adaptation pathways more widely (Haasnoot et al., 2019)
should be considered also in the Venetian context.
Lastly, the recognition of the possible role of compound flood events due to superimposed extreme sea
level drivers shows the potential for improved flood forecasts in Venice, which in turn will allow better
control of the MOSE barriers. Thus, this improved understanding and forecasting of short-term events
will contribute to better long-term adaptation at Venice. With such improved forecasts and greater
confidence in those forecasts, this has the potential to extend the operational range of the MOSE barriers
and its life as an adaptation tool for Venice. This needs to be more fully explored.






### Acknowledgments

The authors thank Federica Braga of ISMAR-CNR for providing the image used in Figure 2a. Scientific
activity by GU and DZ performed in the Research Programme Venezia2021, with the contribution of
the Provveditorato for the Public Works of Veneto, Trentino Alto Adige and Friuli Venezia Giulia,
provided through the concessionary of State Consorzio Venezia Nuova and coordinated by CORILA.
The authors thank Gianfranco Tagliapietra for the historical photo of the flooding of Saint Mark square
on 4th November 1966.

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

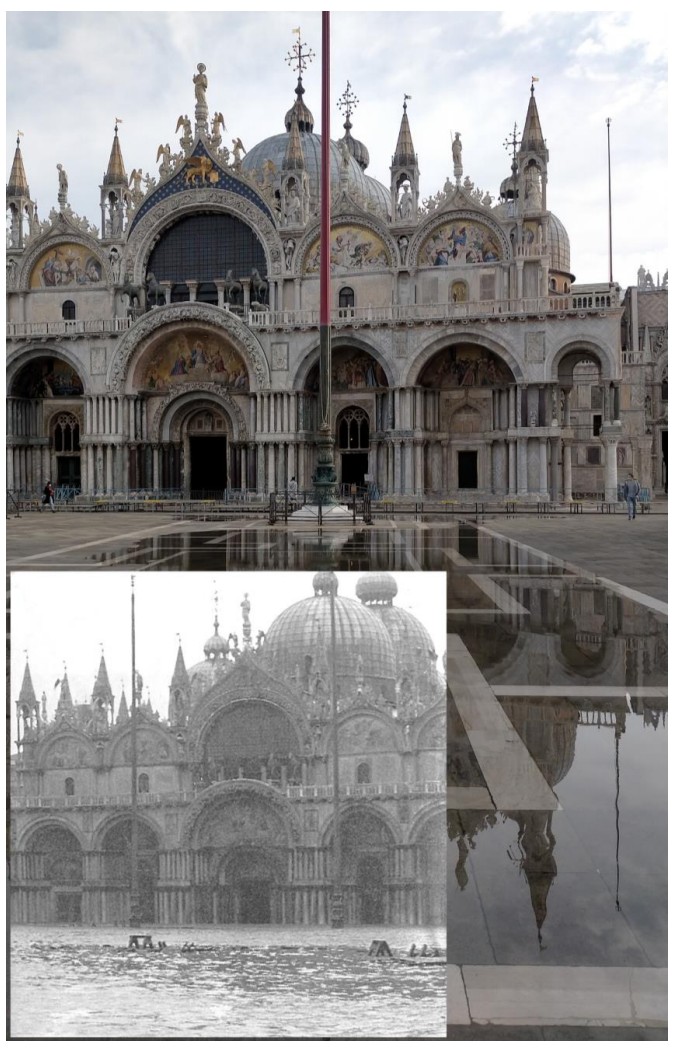


**Figure 1** - Present condition of St.Mark square during partial flooding of its lowest areas (estimated sea level:
80cm) and historical picture close to the time of highest water on the 4 November 1966 flood (black and white
photo courtesy of Gianfranco Tagliapietra)



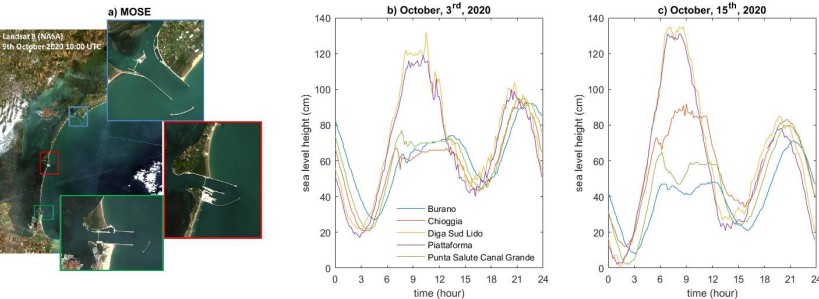

**Figure 2** - Pre-operational closures of the MoSE in October 2020. a) Pseudo-true-colour pan sharpened Landsat
8 OLI imagery acquired on 9 October 2020 showing the Venice Lagoon inlets during a test closure of the MoSE.
Landsat 8 image is available from the U.S. Geological Survey and processed by CNR-ISMAR. b) relative sea
level height anomalies measured on 3 October 2020 by tide gauges located within the Venice lagoon (Burano,
Chioggia, Punta della Salute-Canal Grande) and in the open Adriatic Sea (Diga Sud Lido, Piattaforma CNR),
showing the effect of the MOSE closure on the sea level inside the lagoon. Panel c) is the same as b) but for the
MoSE Closure on 15 October 2020.

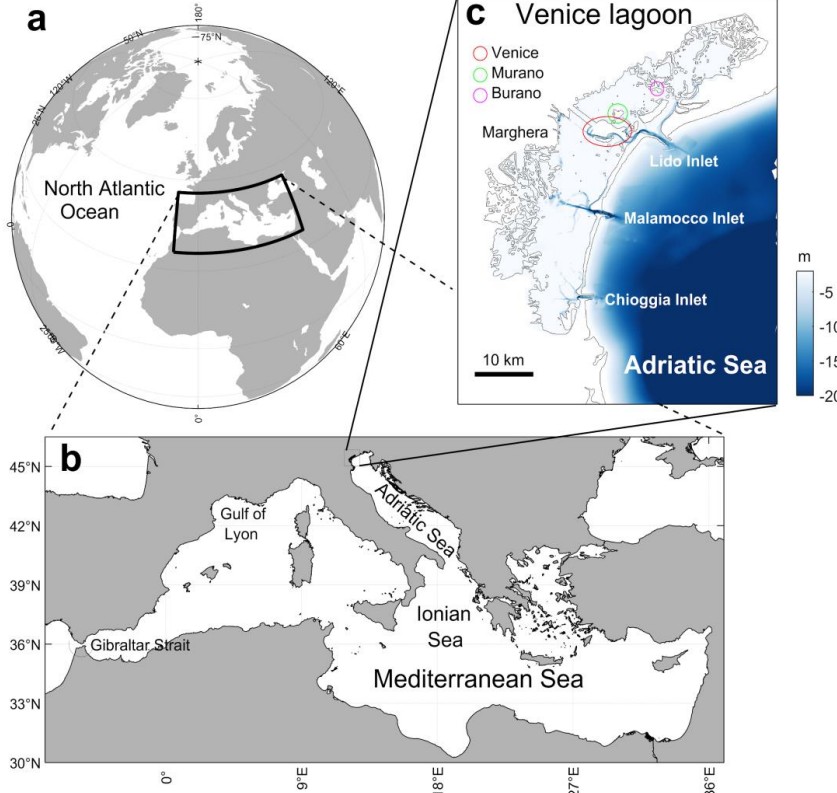

**Figure 3** - The Lagoon of Venice in the global context. (a) The Mediterranean Sea is connected with the North
Atlantic Ocean through the Strait of Gibraltar. (b) The Venice Lagoon is located along the northern coast of the
Adriatic Sea, a sub-basin of the Eastern Mediterranean Sea. (c) The historic center of Venice (indicated) is
located in the center of the Venice Lagoon.

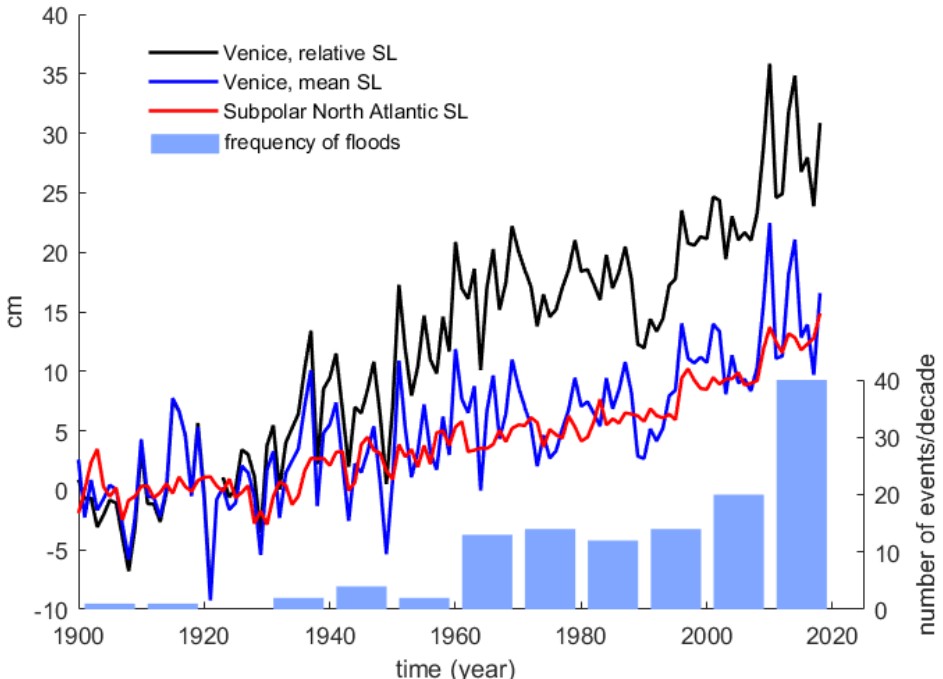

**Figure 4 -** Comparison between historical evolution of average sea level and flooding events in Venice, and the link with larger-scale changes in sea level. Venetian sea level is reported as annual-average relative sea level obtained from measurements by the Punta della Salute tide gauge (black line) and as annual-average mean sea level obtained by removing the local subsidence estimate from the tide gauge data (blue line, Zanchettin et al., 2020). The red line illustrates the evolution of the basin-average sea level for the subpolar North Atlantic estimated by Frederikse et al. (2020). Blue bars show the number of floods exceeding the threshold of 120 cm within each decade.

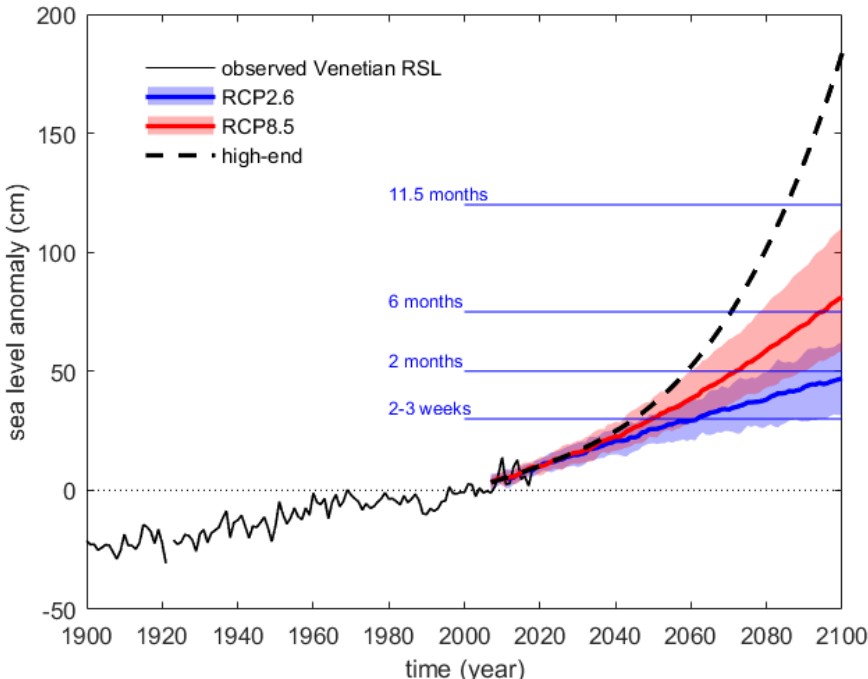

**Figure 5 -** Projected sea level change in Venice in the context of historical observations. Observations are annual-
mean tide gauge relative sea-level height anomalies with respect to the 2000-2007 average. Projections are based
on two reference scenarios of anthropogenic greenhouse gas emission, namely RCP2.6 (low emission scenario)
and RCIP8.5 (strong emission scenario), and a high-end scenario illustrating a plausible evolution  obtained by
combining the highest estimates of all individual contributions to relative sea level rise. The horizontal blue lines
shows the annual persistence of the sea level above the present safeguard level as a function of future relative
mean sea level. These time intervals approximately correspond to the annual duration of the expected closures of
MoSE.