# Peer review of "Introduction to the special issue "Venice flooding and sea level: past evolution, present issues and"

_Natural Hazards and Earth System Sciences, 2020_

## Referee Comment (RC1) · Anonymous Referee #1 · 19 Jan 2021

This short paper represents the introduction for a special issue focused on flooding issues in Venice, in the past, present and future. The special issue comprises 3 papers which are all reviewed separately. The introduction includes a brief summary of the key findings and conclusions of the more detailed technical papers. The goal of such an introductory article is to "set the stage" for the special issue and to provide a short outlined of what the issue comprises. This paper does exactly that and I have no major comments or concerns. The only thing that the authors may want to check (and possibly revise) is the wording in lines 50-51. If the threshold is at 55 cm above mean sea level and the tidal amplitude of the max tide is 50 cm it indicates that only 5 cm surge (or wave setup, or small mean sea level anomaly) are required to push the total

water level beyond the threshold. This is a bit contradictory with the statement that a "moderate" surge is required (which is also not very well defined).

---

## Author Comment (AC1) · 31 Jan 2021

We thank reviewer 1 for these comments. The numerical information in the quoted sentence referring to St. Mark's Square is correct. However, it is, indeed, appropriate rewording it. In the revised text we will explain that "...nowadays even a positive sea level anomaly that is only few centimeters above astronomical high tide (whose level is approximately 50cm) can partially flood it ...".

---

## Referee Comment (RC2) · Anonymous Referee #2 · 11 Mar 2021

I generally would like to follow Referee #1 and thus do not need to repeat the summary or the overall intention of the paper by Lionello et al.; as Referee #1, I also do not have major comments but would like to provide some more specific comments/questions as follows:

Line 49: It would be nice to get more information on the average number of flooding events nowadays against a selected period from the past.

After reading the entire paper, one could also just point to the following sections (and/or to Fig 4, which then becomes Fig 1...) where this is discussed/presented.

Line 55: In my understanding, the event of 1966 should be described as storm surge,

"reaching the highest ever recorded extreme sea levels (ESL)"; the term RSL refers to just on component of the entire ESL, consisting of tide, surge, RSL. I often find it difficult to read papers on storm surges, ESL or MSL changes as the terms used to describe these phenomena differ from author to author. I like the using the terminology proposed by Gregory et al. (2019).

Gregory, J.M., Griffies, S.M., Hughes, C.W. et al. Concepts and Terminology for Sea Level: Mean, Variability and Change, Both Local and Global. Surv Geophys 40, 1251–1289 (2019). https://doi.org/10.1007/s10712-019-09525-z

Line 62: Following the suggestion above, I would use ESL instead of sea levels; if considered relevant, more adjustments are needed throughout the manuscript but these are no longer highlighted here

Line 105: Comma after Century

Line 132: Mixed tidal-regime, semi- or diurnal? Would be nice to know, especially as you refer to the diurnal frequency of tides hereafter.

Line 159: "...of the compound event that led to the exceptional sea level maximum". Compound events (and a specific one) have not been discussed before, somehow confusing the reader. Maybe write "a compound event (discussed in/see XXXX)" and maybe also give some short explanation as e.g. "the joint occurrence of two or more individual hazards...".

Line 231: Still difficult to assess, as all scenarios develop side by side. Based on the article of Hausfather and Peters (2020) I would add "...the world seems heading...".

Fig. 2: Nice, but really small and thus difficult to read. Maybe put the aerial above the two sea level graphs and extend all slightly

---

## Author Comment (AC2) · 19 Apr 2021

**Answers to the comments of the reviewer**

*Here are on the behalf of all authors, the replies (text in Italic) to the comments of Reviewer#2 (text in bold characters)*

**I generally would like to follow Referee #1 and thus do not need to repeat the summary or the overall intention of the paper by Lionello et al.; as Referee #1, I also do not have major comments but would like to provide some more specific comments/questions as follows:**

*We thank the reviewer for these constructive and useful comments. Please, see below our detailed answers to each specific comment.*

**Line 49: It would be nice to get more information on the average number of flooding events nowadays against a selected period from the past. After reading the entire paper, one could also just point to the following sections (and/or to Fig 4, which then becomes Fig 1: : : ) where this is discussed/presented.**

*Thanks for suggesting this.*
*In fact, this information is shown in Figure 4, but it was not highlighted in the text. Figure 4 shows that the number of floods above the 120 cm threshold has increased from 1.6/decade (average frequency during the first half of the 20$^{th}$ century) to 40/decade in the period 2010-2019. Considering the lower 110 cm threshold the number of events has increased from 4.2/decade to 95/decade during the same time spans. This information will be added to the text of this manuscript (hereafter Editorial) and to section 4.1 of the review paper on Extreme floods of Venice in this special issue (hereafter L2020), where discussing the past evolution and recent trends of floods and extreme sea levels in.*

**Line 55: In my understanding, the event of 1966 should be described as storm surge, "reaching the highest ever recorded extreme sea levels (ESL)"; the term RSL refers to just on component of the entire ESL, consisting of tide, surge, RSL. I often find it difficult to read papers on storm surges, ESL or MSL changes as the terms used to describe these phenomena differ from author to author. I like the using the terminology proposed by Gregory et al. (2019).**
**Gregory, J.M., Griffies, S.M., Hughes, C.W. et al. Concepts and Terminology for Sea Level: Mean, Variability and Change, Both Local and Global. Surv Geophys 40, 1251–1289 (2019).**
**https://doi.org/10.1007/s10712-019-09525-z**
**Line 62: Following the suggestion above, I would use ESL instead of sea levels; if considered relevant, more adjustments are needed throughout the manuscript but these are no longer highlighted here**

*We consider together these two comments, because they are related.*
*First, thanks for pointing at the Gregory et al. (2019) paper (hereafter G2019). We agree it provides an excellent reference for the terminology to be used when discussing sea level. However, two issues prevent us following strictly G2019 in our special issue.*
*The first issue is the proper characterization of extremes for the Venice sea level, which are actually extremes of the local instantaneous thickness of the ocean, $\widetilde{H} = \tilde{\eta} - \widetilde{F}$, where $\tilde{\eta}$ and $\widetilde{F}$ are the instantaneous sea surface and bottom level, respectively (here the meaning of the symbols is the same as in G2019). Both $\tilde{\eta}$ and $\widetilde{F}$ are measured with respect to a common reference (which could be the reference ellipsoid, the geoid G, or a geocentric reference frame). On the contrary, in G2019 sea surface height extremes are defined as exceptionally high values of the geodetic height of the sea surface above the reference ellipsoid $\tilde{\eta} - G$. Therefore, the characteristics of the extremes practically relevant for the flood of Venice do not correspond to the definition of sea surface height extremes in G2019. We could refer to the extremes of $\widetilde{H}$ using the term "extreme thickness", but this sounds to us a clumsy and not effective terminology. We suggest to use the terminology "water height" extremes or high-end values.*

*We note that a similar issue on terminology arose also for the review article on mean sea level trends (Zanchettin et al., 2020, this special issue, hereafter Z2020), and further discussion is available in our responses to the reviewers in the open discussion of that manuscript (see: [https://nhess.copernicus.org/preprints/nhess-2020-351/](https://nhess.copernicus.org/preprints/nhess-2020-351/)). In short, as Z2020 deals mainly with mean and relative sea level issues, we could adopt there the G2019 terminology with a few exceptions related to the Earth's gravity, rotation and deformation components, which are clarified in the text.*
*We agree with the reviewer that a general revision of the terminology is needed in this introduction and L2020, and we will implement it in the revised versions, with an explanation of the terms used and how they differ from G2019.*

*The second issue is the definition of surge that according to G2019 is "The elevation or depression of the sea surface with respect to the predicted tide during a storm". Actually, as it is discussed in L2020, the sea surface elevation with respect to the tide results from contributions with different time scale. Some contributions have a time scale that is much longer than the typical duration of a storm, O(24 hours), some shorter, namely the meteo-tsunamis whose duration is O(1hour), and basin wide seiches have their own periods of approximately 11 and 22 hours. Therefore, the G2019's definition of surge is not the best option for the analysis of sea level extremes in the North Adriatic and Venice, though we acknowledge that it is applicable to a wide range of situations that are simpler than the North Adriatic Sea. This discussion will be added to this Editorial and will be further expanded in the review article L2020.*
*Considering specifically the November 1966 event, the surge component (as defined in L2020) is the main contribution to the extreme water height value, but in other cases (e.g. the event of November 2019) other contributions have had comparable magnitude.*

**Line 105: Comma after Century**
*Thanks*

**Line 132: Mixed tidal-regime, semi- or diurnal? Would be nice to know, especially as you refer to the diurnal frequency of tides hereafter.**

*The tidal regime is a mixed semidiurnal cycle with two high and two low tide levels of different height every day. There are 7 components with amplitude above 1 centimeter and only 3 above 10 centimeters, with semidiurnal M2 and S2, and diurnal K1 providing the largest contributions. The values of M2, S2 and K1 are approximately 23, 14, 16 centimeters, respectively, both outside the lagoon and at the Venice historical tide gauge station. This information will be added below line 132.*

**Line 159: ": : :of the compound event that led to the exceptional sea level maximum". Compound events (and a specific one) have not been discussed before, somehow confusing the reader. Maybe write "a compound event (discussed in/see XXXX)" and maybe also give some short explanation as e.g. "the joint occurrence of two or more individual hazards:**

*We will swap the order of paragraphs beginning at line 159 and 165. The latter describes the relevance of compound events. Extreme water heights are the result of different factors, often playing comparable roles. The review article L2020 lists and describes them: astronomical tides, seiches, storm surges, meteo-tsunamis, long planetary atmospheric waves, with a background trend produced by relative sea-level rise. Compound events occurs when their superposition produce an extreme water height, though individual factors have a non-exceptional height. This discussion will be added to the paragraph.*

Line 231: Still difficult to assess, as all scenarios develop side by side. Based on the article of Hausfather and Peters (2020) I would add " the world seems heading".

*We agree and have changed the text accordingly.*

Fig. 2: Nice, but really small and thus difficult to read. Maybe put the aerial above the two sea level graphs and extend all slightly

*Thanks for the suggestion.*

---

## Author Response (AR1)

Dear Editor:

On the behalf of all authors, I thank the reviewers for their positive evaluation and constructive comments. Please, find below the description of the changes implemented in the article following their suggestions. Reviewer's comments are in bold characters, authors' responses are in slant characters, excerpts from the manuscript are in red characters

After the submission of this introduction, the revisions of the three review articles that are cited in this introduction have produced three minor changes, beside those suggested by the reviewers.
They are:

- The addition to the abstract of the sentence *"… relative sea level rise is the key factor determining the future growth of the flood hazard, so that the present defence strategy is likely to become inadequate within this century under a high emission scenario"*
- The event of 8 December 2020, which occurred after the submission of the review articles is mentioned in the text "*The need to improve the operational forecasting system has been further shown by the flooding of the city on 8 December 2020, when the MoSE was not operated, in spite of being available, because the forecast underestimated the height of the water level.*"
- We have better detailed the possible consequences of projected RSL on the future duration of the closures of MoSE ( the movable barriers at the lagoon inlets):*" Fig. 5 reports the RSL thresholds for the closures based on the consensus between Lionello (2012) and Umgiesser (2020) and it shows that the period of closure will grow at a rate controlled by RSL rise. Closing of the inlets for three weeks per year is unlikely before the 2040's, but virtually certain before the end of this century, even under a low emission scenario (RCP2.6). Two months closures per year are unlikely before the late 2050's even under a high-end emission scenario (RCP8.5). However, they become virtually certain in the late 2080's under a high emission scenario and about as likely as not before the end of this century for a low emission scenario. Note that a six-month closure per year (which can be used as criterion for considering the present defence strategy to be inadequate and requiring new additional actions) is likely to occur before the end of this century under a high emission scenario.*"

Note, that following the comment on the terminology of reviewer #2 and the consequent changes that have been implemented in the review article on sea-level extremes (here referred to as L2021), "water height" is used in the text of this introduction.

Concerning the copyright and authorship of the background panel of figure 1, the author has been identified. He has sent an email with his consent to the use of the picture and is acknowledged in the caption.

Best regards

Piero Lionello

**Answers to the comments of reviewer #1**

**This short paper represents the introduction for a special issue focused on flooding issues in Venice, in the past, present and future. The special issue comprises 3 papers which are all reviewed separately. The introduction includes a brief summary of the key findings and conclusions of the more detailed technical papers. The goal of such an introductory article is to "set the stage" for the special issue and to provide a short outlined of what the issue comprises. This paper does exactly that and I have no major comments or concerns. The only thing that the authors may want to check (and possibly revise) is the wording in lines 50-51. If the threshold is at 55 cm above mean sea level and the tidal amplitude of the max tide is 50 cm it indicates that only 5 cm surge (or wave setup, or small mean sea level anomaly) are required to push the**

**total water level beyond the threshold. This is a bit contradictory with the statement that a "moderate" surge is required (which is also not very well defined).**

*Thank you for your positive evaluation of this article. The numerical information in the quoted sentence referring to St.Mark's Square is correct. However, a minor rewording stating that "… nowadays a positive water height anomaly (see section 3) that is only a few centimeters above astronomical high tide (whose amplitude is about 50cm) can flood it." is, indeed, appropriate.*

**Answers to the comments of reviewer #2**

*Here are on the behalf of all authors, the replies (text in Italic) to the comments of Reviewer#2 (text in bold characters)*

**I generally would like to follow Referee #1 and thus do not need to repeat the summary or the overall intention of the paper by Lionello et al.; as Referee #1, I also do not have major comments but would like to provide some more specific comments/questions as follows:**

*We thank the reviewer for these constructive and useful comments. Please, see below our detailed answers to each specific comment.*

**Line 49: It would be nice to get more information on the average number of flooding events nowadays against a selected period from the past. After reading the entire paper, one could also just point to the following sections (and/or to Fig 4, which then becomes Fig 1: : : ) where this is discussed/presented.**

*Thanks for suggesting this.*
*In fact, this information is shown in Figure 4, but it was not highlighted in the text. Figure 4 shows that the number of floods above the 120 cm threshold has increased from 1.6/decade (average frequency during the first half of the 20$^{th}$ century) to 40/decade in the period 2010-2019. The following sentence has been added to the text of this manuscript in section 4 when commenting Fig.4: "Figure 4 summarizes these results by showing the RSL rise in Venice and the corresponding increase of frequency of water-height maxima above 120 cm, which has increased from less than two events per decade (average frequency during the first half of the 20th century) to 40 events in the last decade (2010-2019). Considering a lower (110 cm) water-height threshold, the number of events has increased from 4.2/decade to 95/decade". More information about this can be found in section 4.1 of the review paper on Extreme floods of Venice in this special issue (hereafter L2021), where discussing the past evolution and recent trends of floods and extreme sea levels in.*

**Line 55: In my understanding, the event of 1966 should be described as storm surge, "reaching the highest ever recorded extreme sea levels (ESL)"; the term RSL refers to just on component of the entire ESL, consisting of tide, surge, RSL. I often find it difficult to read papers on storm surges, ESL or MSL changes as the terms used to describe these phenomena differ from author to author. I like the using the terminology proposed by Gregory et al. (2019).**
**Gregory, J.M., Griffies, S.M., Hughes, C.W. et al. Concepts and Terminology for Sea Level: Mean, Variability and Change, Both Local and Global. Surv Geophys 40, 1251–1289 (2019).**
**https://doi.org/10.1007/s10712-019-09525-z**
**Line 62: Following the suggestion above, I would use ESL instead of sea levels; if considered relevant, more adjustments are needed throughout the manuscript but these are no longer highlighted here**

*We consider together these two comments, because they are related.*

*First, thanks for pointing at the Gregory et al. (2019) paper (hereafter G2019). We agree it provides an excellent reference for the terminology to be used when discussing sea level. However, two issues prevent us following strictly G2019 in our special issue.*

*The first issue is the proper characterization of extremes for the Venice sea level, which are actually extremes of the local instantaneous thickness of the ocean, $\tilde{H} = \tilde{\eta} - \tilde{F}$, where $\tilde{\eta}$ and $\tilde{F}$ are the instantaneous sea surface and bottom level, respectively (here the meaning of the symbols is the same as in G2019). Both $\tilde{\eta}$ and $\tilde{F}$ are measured with respect to a common reference (which could be the reference ellipsoid, the geoid G, or a geocentric reference frame). On the contrary, in G2019 sea surface height extremes are defined as exceptionally high values of the geodetic height of the sea surface above the reference ellipsoid $\tilde{\eta} - G$. Therefore, the characteristics of the extremes practically relevant for the flood of Venice do not correspond to the definition of sea surface height extremes in G2019. We could refer to the extremes of $\tilde{H}$ using the term "extreme thickness", but this sounds to us a clumsy and not effective terminology. We have decided to use the terminology "water height" extremes or high-end values.*

*We agree with the reviewer that a general revision of the terminology was needed in this introduction and in L2021, and we implemented it in the revised versions, with an explanation of the terms used and why they differ from G2019.*

*The second issue is the definition of surge that according to G2019 is "The elevation or depression of the sea surface with respect to the predicted tide during a storm". Actually, as it is discussed in L2021, the sea surface elevation with respect to the tide results from contributions with different time scales. Some contributions have a time scale that is much longer than the typical duration of a storm, typically O(24 hours) in the Adriatic Sea, some shorter, namely the meteo-tsunamis whose duration is O(1hour), and basin wide seiches have their own periods of approximately 11 and 22 hours. Therefore, the G2019's definition of surge is not the best option for the analysis of sea level extremes in the North Adriatic and Venice, though we acknowledge that it is applicable to a wide range of situations that are simpler than the North Adriatic Sea.*

*The following paragraph has been added at the beginning of section 3* *"The floods of Venice are associated with the positive anomalies of the water height, defined as the difference between the instantaneous sea level and the bottom level. The term "water height" is introduced because considering only sea level does not account for the fundamental role that the local vertical land motion (subsidence) has and will continue to have in the increased frequency of floods. The contributions leading to large water height anomalies are meteorological surges, seiches, tides, seasonal-to-decadal sea level variability and long term RSL changes. The meteorological surges result from three different contributions characterized by different time scales: surges produced by planetary atmospheric waves (PAW surges), with duration from 10 days to 100 days, storm surges produced by mid-latitude cyclones with time scales of a few days, meteotsunamis and surges produced by mesoscale systems (with a short duration of a few hours). The characteristics of the different contributions and the criteria for their distinction are explained in section 2.1 and 2.2 of L2021. Note that this terminology differs from Gregory et al. (2019) in that the term "water height" is introduced and the surge is distinguished by three components, reserving the term "storm surge" for the component produced by the passage of a cyclone."*

*Considering specifically the November 1966 event, the surge component (as defined in L2021) is the main contribution to the extreme water height value, but in other cases (e.g. the event of November 2019) other contributions have had comparable magnitude. This is explained in the initial paragraph (minor changes have been implemented for increasing its clarity) of section 4:*

*"The important potential role of compound events (resulting from the superposition of different contributions) for causing extreme sea levels emerges clearly from L2021. Many past studies concentrated on the storm surge contribution, which was the determinant contribution for the 4 November 1966 and the 19 October 2018 events, and on the need for a precise prediction of its timing in relation to the phase of the astronomical tide and pre-existing seiches. However, the presence of other factors can determine extreme sea-level events when they act constructively, namely planetary atmospheric wave surges and meteotsunamis, even if their individual magnitude*

*is not exceptionally large (L2021), as it was apparent in the recent 12 November 2019 event (Ferrarin et al., 2021). This poses a great challenge to the prediction of extreme sea levels (U2021) and the management of MoSE. Further, historic floods show large interdecadal and interannual fluctuations, whose dynamics are not sufficiently understood (L2021), preventing reliable seasonal predictions."*

**Line 105: Comma after Century**
*Thanks*

**Line 132: Mixed tidal-regime, semi- or diurnal? Would be nice to know, especially as you refer to the diurnal frequency of tides hereafter.**

The following text has been added to section 2, 3rd paragraph: "*The tidal regime is a mixed semidiurnal cycle with a tidal range of more than 1 m at spring tide and only three components above 10 cm, with the semidiurnal M2 and S2, and the diurnal K1 providing the largest contributions (23, 14, 16 cm, respectively) both outside the lagoon inlets and in the city center (Polli, 1952; Ferrarin et al. , 2015)*"

**Line 159: ": of the compound event that led to the exceptional sea level maximum". Compound events (and a specific one) have not been discussed before, somehow confusing the reader. Maybe write "a compound event (discussed in/see XXXX)" and maybe also give some short explanation as e.g. "the joint occurrence of two or more individual hazards:**

*The text has been rephrased. The new first paragraph at the beginning of section 3 provides a basic description of the different contributions and the second paragraph introduces the compound event resulting from their their superposition. In section 4, the paragraph formerly at line 165 has been moved at the beginning of the section and a sentence referring explicitly to section 3 has been added. Extreme water heights are the result of different factors, often playing comparable roles: astronomical tides, seiches, storm surges, meteo-tsunamis, long planetary atmospheric waves, with a background trend produced by relative sea-level rise. Compound events occurs when their superposition produce an extreme water height, though individual factors have a non-exceptional height. We think the new text in section 3 avoids the potential confusion highlighted by Reviewer #2.*

**Line 231: Still difficult to assess, as all scenarios develop side by side. Based on the article of Hausfather and Peters (2020) I would add " the world seems heading".**

*We agree that there is an uncertainty and have changed the text accordingly:"Currently, it is not clear whether the world is heading towards emissions more comparable to RCP4.5 or RCP6.0 (Hausfather and Peters, 2020), rather than to RCP8.5 (Schwalm et al., 2020)".*

**Fig. 2: Nice, but really small and thus difficult to read. Maybe put the aerial above the two sea level graphs and extend all slightly**

*Thanks for the suggestion.*